

# A survey of fishes associated with Hawaiian deep-water *Halimeda kanaloana* (Bryopsidales: Halimedaceae) and *Avrainvillea* sp. (Bryopsidales: Udoteaceae) meadows

Ross C. Langston[1] and Heather L. Spalding[2]

[1] Department of Natural Sciences, University of Hawai'i- Windward Community College, Kāne'ohe, HI, USA
[2] Department of Botany, University of Hawai'i at Mānoa, Honolulu, HI, USA

## ABSTRACT

The invasive macroalgal species *Avrainvillea* sp. and native species *Halimeda kanaloana* form expansive meadows that extend to depths of 80 m or more in the waters off of O'ahu and Maui, respectively. Despite their wide depth distribution, comparatively little is known about the biota associated with these macroalgal species. Our primary goals were to provide baseline information on the fish fauna associated with these deep-water macroalgal meadows and to compare the abundance and diversity of fishes between the meadow interior and sandy perimeters. Because both species form structurally complex three-dimensional canopies, we hypothesized that they would support a greater abundance and diversity of fishes when compared to surrounding sandy areas. We surveyed the fish fauna associated with these meadows using visual surveys and collections made with clove-oil anesthetic. Using these techniques, we recorded a total of 49 species from 25 families for *H. kanaloana* meadows and surrounding sandy areas, and 28 species from 19 families for *Avrainvillea* sp. habitats. Percent endemism was 28.6% and 10.7%, respectively. Wrasses (Family Labridae) were the most speciose taxon in both habitats (11 and six species, respectively), followed by gobies for *H. kanaloana* (six species). The wrasse *Oxycheilinus bimaculatus* and cardinalfish *Apogonichthys perdix* were the most frequently-occurring species within the *H. kanaloana* and *Avrainvillea* canopies, respectively. Obligate herbivores and food-fish species were rare in both habitats. Surprisingly, the density and abundance of small epibenthic fishes were greater in open sand than in the meadow canopy. In addition, species richness was also higher in open sand for *Avrainvillea* sp. We hypothesize that the dense holdfasts and rhizoids present within the meadow canopy may impede benthic-dwelling or bioturbator species, which accounted for 86% and 57% of individuals collected in sand adjacent to *H. kanaloana* and *Avrainvillea* sp. habitats, respectively. Of the 65 unique species recorded in this study, 16 (25%) were detected in clove oil stations alone, illustrating the utility of clove-oil anesthetic in assessing the diversity and abundance of small-bodied epibenthic fishes.

Corresponding author
Ross C. Langston,
langston@hawaii.edu

## INTRODUCTION

Macroalgal meadows constitute important habitats for reef- and nearshore fish species. Many are important grazing areas for herbivorous fishes (*Lobel & Ogden, 1981*) and may also serve as spawning sites for recreationally important food fishes such as parrotfish and wrasses (*Colin & Bell, 1991*). Macroalgae may also serve a key role in ontogenetic habitat shifts in post-settlement fish (*Eggleston, 1995*; *Dahlgren & Eggleston, 2000*). Relative to surrounding habitats, which are often sandy and of low-relief, macroalgal meadows constitute highly-complex and rugose habitats which may afford protection to juvenile and small-bodied reef fish species alike.

Until recently, most detailed studies of reef fish diversity have been limited to 30 m or less (e.g., *Randall, 1998*; *Greenfield, 2003*), which is the generally accepted limit for conventional SCUBA diving. However, with the recent advent of Closed Circuit Rebreathers (CCR) and mixed-gas diving technology, properly-trained researchers are now able to work at depths of 100 m or more (e.g., *Pyle, 2000*; *Lesser, Slattery & Leichter, 2009*; *Khang et al., 2010*; *Kane, Kosaki & Wagner, 2014*; *Kosaki et al., 2016*; *Simon et al., 2016*). Recent work has focused on the mesophotic zone, which extends from 30 to 150 m  (*Hinderstein et al., 2010*). Much of this work has focused on coral reef habitats and their associated fauna. Few studies have investigated the fauna of mesophotic macroalgae, despite the fact that several meadow-forming species occur at depths of 50 m or more in tropical and subtropical waters (*Huisman, Abbott & Smith, 2007*; *Spalding, 2012*; *Pyle et al., 2016*).

In this paper, we describe the fish fauna associated with two deep-water macroalgal species, *Avrainvillea* sp. and *Halimeda kanaloana*, from Hawaiian waters. *Avrainvillea* sp. and *H. kanaloana* are siphonous green macroalgae which form predominantly monospecific meadows over large areas of sandy substrate from shallow (1 m) to deep (>80 m) waters in the Main Hawaiian Islands (*Spalding, 2012*). *Halimeda kanaloana* is a calcified alga native to Hawai'i with multiple branched axes up to 30 cm in height (*Verbruggen et al., 2006*). *Avrainvillea* sp., an invasive species that was previously misidentified as *A. amadelpha* (*Wade, Tang & Sherwood, 2015*),  forms dense, mat-like beds with bladed canopies approximately 10 cm in height (*Spalding, 2012*). *Avrainvillea* sp. first appeared off Kahe Point, O'ahu in 1981, and subsequently spread to Maunalua Bay, O'ahu (*Brostoff, 1989*), where it outcompeted native algae and seagrasses (*Peyton, 2009*; *Abbott & Huisman, 2004*). Both species have been reported to support a greater diversity of epibenthic or infaunal invertebrates when compared to surrounding sandy habitats (*Fukunaga, 2008*; *Magalhaes & Bailey-Brock, 2014*); however, little is currently known about the role these assemblages may play in the creation/loss of unique habitat for fishes. Because *Avrainvillea* sp. and *H. kanaloana* occupy similar habitats in Hawai'i (sandy substrate in moderate to low wave environments), they offer an opportunity to determine the effects of canopy type on the composition of the associated fish fauna.

Since little information is currently available for these habitats, our primary goal was to provide baseline information on the fish fauna associated these meadows. As part of this goal, we sought to document the most common and abundant species in both habitats. We also sought to identify any commercially- or recreationally important fish

species so that resource managers can determine if these meadows merit additional study or protection. Based on surveys from shallow-water meadows (e.g., *Lobel & Ogden, 1981*; *Francini-Filho et al., 2010*), we hypothesized that deepwater *H. kanaloana* and *Avrainvillea* sp. meadows might be important feeding areas for herbivorous species such as surgeonfishes or parrotfishes, many of which are prized food-fish species in Hawai'i. In addition, we also attempted to calculate the level of endemism in both macroalgal habitats. Given that deep reefs in the Northwestern Hawaiian Islands (NWHI) support a greater number of endemic species than their shallow-water counterparts (*Kane, Kosaki & Wagner, 2014*), we hypothesized that deep water macroalgal meadows might likewise support a greater proportion of endemic species than shallow habitats in the Main Hawaiian Islands (MHI).

Our final goal was to compare the abundance and diversity of small-bodied epibenthic fishes between open sand and meadow canopy subhabitats. Because both macroalgal species form structurally complex three-dimensional canopies, we hypothesized that they would support a greater abundance and diversity of fishes when compared to surrounding sandy habitats, which are typically of low relief and complexity. This hypothesis is supported by the work of *Chittaro (2004)*, who found that fish abundance and species richness in Tague Bay, St. Croix were positively correlated with *H. incrassata* habitats and negatively correlated with open sand and pavement. Likewise, *Ornellas & Coutinho (1998)* found that fish diversity in sublittoral areas of Cabo Frio Island (Brazil) was greater in *Sargassum furcatum* beds than in surrounding sandy habitats. Thus, it seems likely that *H. kanaloana* and *Avrainvillea* sp. meadows should support a greater diversity and abundance of fishes than adjacent sandy areas.

## MATERIALS AND METHODS

We utilized CCR technical diving to survey fish assemblages during a total of 20 dives (four in *Avrainvillea* sp. habitats and 16 in *H. kanaloana* habitats). *Avrainvillea* sp. habitats were surveyed at a single site off west O'ahu and *H. kanaloana* habitats were surveyed at three sites off of south and west Maui (Fig. 1). All surveys were conducted between June 14th, 2005 and June 12th, 2006. Initial surveys consisted of visual censuses supplemented by collections made with pole spears. All other surveys were conducted using tandem visual surveys combined with collections using clove-oil anesthetic in order to better assess the numbers of small epibenthic fishes (Fig. 2). For these collections, an anesthetic solution of 10% clove oil in 90% ethanol was aspirated from a squirt bottle beneath a 1.5 m weighted plastic tarp. The tarp was placed haphazardly over either sand or canopy. The solution was allowed to work for a period of 10 min, during which the diver conducted visual- and photographic surveys of larger-bodied fish species (see below). After the 10 min had elapsed, the divers removed the weighted tarp and used fine-mesh nets to collect the fishes, which were photographed and preserved in 10% formalin for subsequent identification. Because fishes caught in the clove oil stations were collected from a known area (1.5 m), we were able to compare the species richness and abundance (average number of species and individuals per 1.5 m collection) between sand and canopy sub-habitats.

Conventional transect-based visual surveys were impractical given the limited bottom-time, mobility, and task-load of diver teams. Instead, large-bodied fishes were surveyed
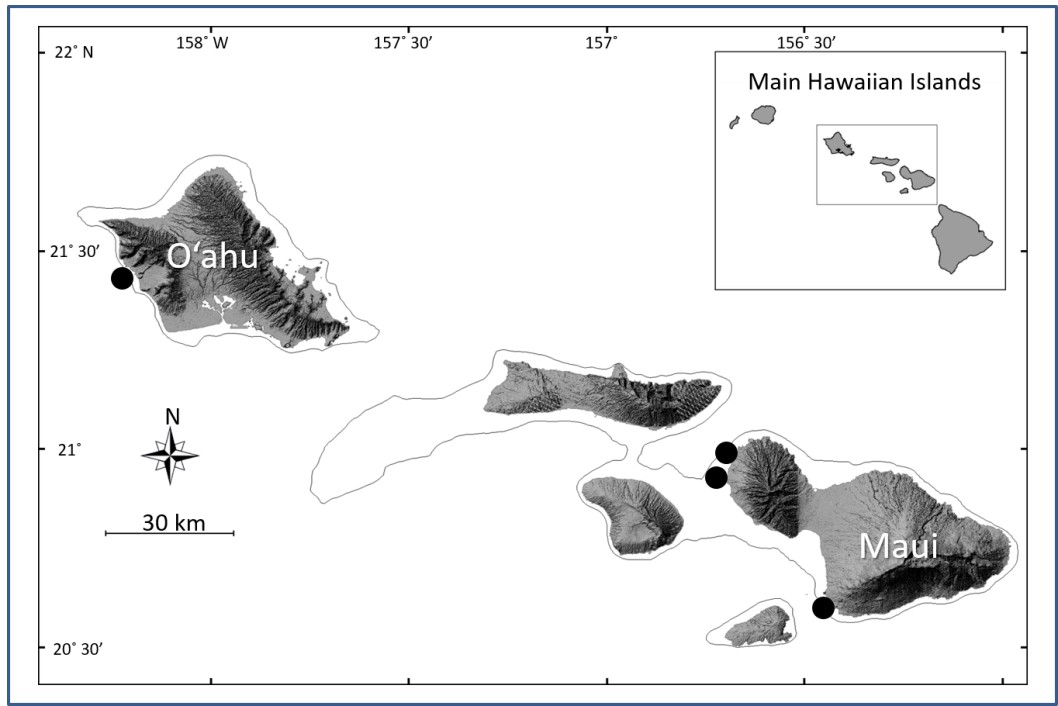

**Figure 1** **Map of the Main Hawaiian Islands showing the locations of the four survey sites.** *Avrainvillea* sp. meadows were surveyed off west Oʻahu. *Halimeda kanaloana* meadows were surveyed off south and west Maui.

using a stationary point count (SPC) in which a diver recorded all fishes that resided or passed through a visually estimated 10 m cylinder centered on the diver's location (the weighted tarp). The survey time for each SPC was typically 10 min. Additional species were collected or surveyed via opportunistic spearfishing and photographs. All fish collections were performed in accordance with a University of Hawaiʻi Institutional Animal Care and Use Protocol (# 06-058). Permission to use clove oil anesthetic was granted by the Hawaiʻi Department of Land and Natural Resources (Permit #s PRO 2006-28 and PRO 2007-47).

## Data analysis

All specimens were identified to species or lowest-possible taxon and classified as endemic (restricted to the Hawaiian Islands, Midway, and Johnston Atoll) or non-endemic using available references (e.g., *Randall, 2007*; *Mundy, 2005*). For both clove oil collections and visual surveys, we recorded the abundance ($N$) of each species as well as the habitat (meadow-forming species) and sub-habitat (canopy or sand & rubble) where each species was observed or collected. We used this information to calculate the species richness ($S$) and a Shannon–Wiener diversity index (log e; $H'$) for each sub-habitat. We calculated percentage occurrence (%Occ) as the proportion of collections made within each sub-habitat in which a species was recorded, and percentage relative abundance (%RA) as the number of individuals of a species recorded within a sub-habitat divided by the total number of individuals that were collected or observed in that sub-habitat. We follow *Randall (1996)* in calculating the percent endemism by dividing the number of endemic

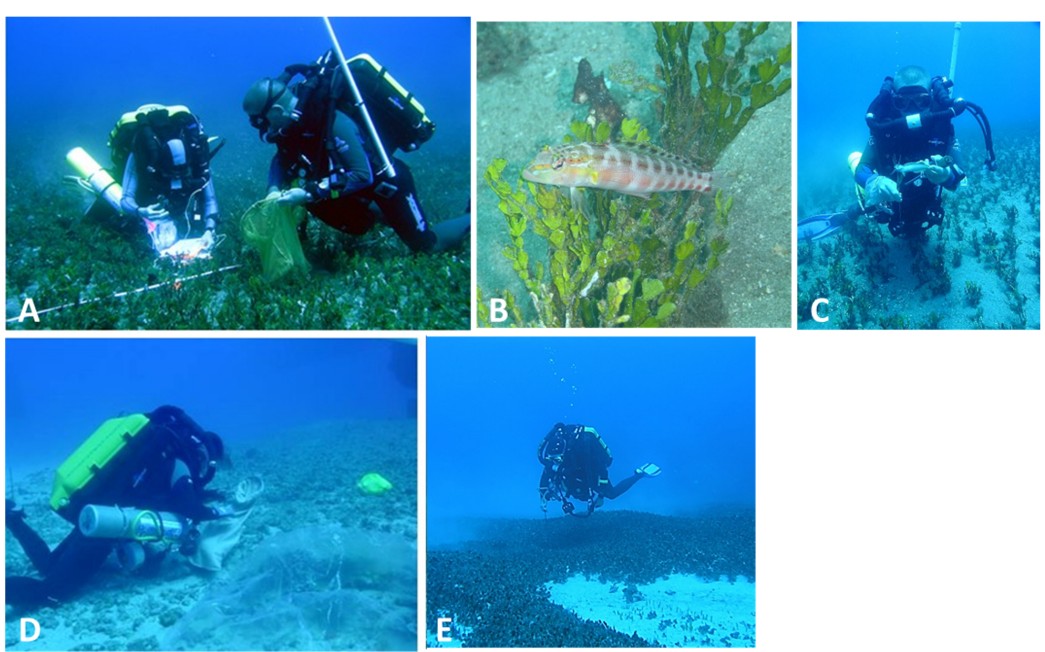

**Figure 2** **Survey techniques used in *Halimeda kanaloana* (A–D) and *Avrainvillea* sp. meadows** Large-bodied species were surveyed visually (A) or photographically (B). Unidentified species were collected with small spears (C). Small-bodied epibenthic fishes were surveyed by injecting a clove-oil solution under a 1.5 m weighted tarp (D). The anesthetized fishes were collected with fine-mesh nets and preserved for subsequent identification. Diver conducting a visual survey of *Avrainvillea* sp. meadow (E). Note the denser canopy.

species by the total number of species present in a particular habitat. We also used available references (*Randall, 1996*; *Randall, 2007*) and personal observations to identify species that reside-upon or feed within the sediment as benthic associates or bioturbators (B) in order to determine if abundance of these species differed between sub-habitats. For clove oil collections, we estimated the density (D) of fishes collected by dividing the number of individuals of each species by the total area of the substrate that was sampled. We compared the median abundance ($N_x$), density ($D_x$), and species richness ($S_x$) within each sub-habitat using the Mood Median Test, as the resulting data did not meet the assumptions necessary to use parametric statistical tests.

## RESULTS

We conducted a total of 14 visual surveys and 51 tandem surveys (visual surveys + collections using clove oil anesthetic) in *Avrainvillea* sp. and *H. kanaloana* habitats (Table 1).

### Percent occurrence and relative abundance of fishes in *H. kanaloana* meadows

A total of 49 species from 25 families were recorded from *H. kanaloana* meadows and surrounding sandy areas (Table 2). Overall species richness and diversity were

**Table 1  Summary of visual surveys and tandem (visual+clove-oil) collections by habitat type and location.**

| Survey Type | Avrainvillea sp. | | Halimeda kanaloana | |
| --- | --- | --- | --- | --- |
| | Canopy | Sand/Rubble | Canopy | Sand/Rubble |
| Visual Only | 2 | 0 | 8 | 4 |
| Clove Oil + Visual | 8 | 3 | 28 | 12 |
| Depth Range | 37–47 m | | 11–40 m | |
| Dates | 10/26/2005–6/12/2006 | | 5/15/06–5/23/06 | |
| Collection Locations | | | | |
| | Makaha, O'ahu 21°26′50.22″N 158°12′42.24″W | | Kahekili Beach Park, Maui 20°56′14.33″N 156° 41′40.54″W Honokowai Beach Park, Maui 20°57′21.00″N 156°41′19.96″W Makena Beach Park, Maui 20°37′42.35″N 156°27′15.04″W | |

nominally greater in canopy ($S = 31$ species, $H' = 2.523$) than in open sand ($S = 29$ species, $H' = 2.064$). Wrasses (Family Labridae) were the most speciose taxon within both sub-habitats followed by Gobies. Wrasses were also the most abundant taxon within meadow canopy, accounting for 54.8% of individuals collected or observed. In contrast, gobies comprised only 15.9% of individuals surveyed in the *H. kanaloana* canopy. The most abundant and frequently-occurring species within meadow canopy was the Two-spot wrasse, *Oxycheilinus bimaculatus*. Other commonly-occurring species included the goby, *Gnatholepis* spp., and the wrasses *Pseudojuloides cerasinus* and *Cymolutes praetextatus*. The latter species was previously unknown east of the Marshall Islands, and thus constitutes a new record for Hawai'i (see *Randall, Langston & Severns, 2006*).

In contrast to *H. kanaloana* canopy, adjacent sandy areas were numerically dominated by goby species (Total %RA = 56.9%), whereas wrasses only accounted for 4.9% of individuals surveyed within this sub-habitat. The three most commonly-occurring species were gobies: *Gnatholepis* spp., *Opua nephodes*, and *Psilogobius mainlandi*.

### Percent occurrence and relative abundance of fishes in *Avrainvillea* sp. meadows

A total of 28 species from 19 families were recorded from *Avrainvillea* sp. meadows and adjacent sandy habitats (Table 3). Overall species richness and diversity were nominally greater in open sand ($S = 19$ species, $H' = 2.74$) than in meadow canopy ($S = 13$ species, $H' = 1.871$). Wrasses were the most speciose and abundant taxon in both sub-habitats. All other taxa were represented by two species or fewer.

The most commonly occurring species within meadow canopy was the cardinalfish *Apogonichthys perdix*. This species was recorded only from clove oil collections and was never observed in visual surveys. Other common species found in *Avrainvillea* sp. canopy include the wrasses *O. bimaculatus* and *Pseudocheilinus evanidus*. The two most abundant species were the unicorn fish *Naso caesius* and Hawaiian Flame Wrasse, *Cirrhilabrus jordani*, however, these species were recorded from a single collection each.

Only three collections were made in sandy habitats adjacent to *Avrainvillea* sp. meadows. Overall, wrasses were the most abundant taxon in open sand (%RA = 23.3%). The most

**Table 2 Checklist of fishes associated with deep-water *Halimeda kanaloana* meadows based on clove oil collections and visual surveys.** Sub-habitats are listed as Meadow Canopy (directly within vegetation) and Sand/Rubble (blow-outs and sandy & meadow perimeters). Endemic species are indicated by ''E'' whereas those which rest directly upon- or feed within the substrate are indicated by ''B''. CO(N) and V(N) indicate the numbers of each species collected or surveyed in clove oil collections or visual surveys, respectively. All other abbreviations are described in the methods.

| Species | E, B | Meadow canopy | | | | Sand/Rubble | | | |
|---|---|---|---|---|---|---|---|---|---|
| | | CO(N) | V(N) | % Occ | % RA | CO(N) | V(N) | % Occ | % RA |
| Myliobatidae | | | | | | | | | |
| *Aetobatus narinari* | | | | | | | 1 | 6.3 | 0.2 |
| *Manta birostris* | | | | | | | 1 | 6.3 | 0.2 |
| Congridae | | | | | | | | | |
| *Conger cinereus* | | 1 | | 2.8 | 0.3 | | | | |
| Synodontidae | | | | | | | | | |
| *Synodus spp.* | B | | 4 | 8.3 | 1.2 | | | | |
| Aulostomidae | | | | | | | | | |
| *Aulostomus chinensis* | | | 2 | 2.8 | 0.6 | | | | |
| Fistulariidae | | | | | | | | | |
| *Fistularia commersonii* | | | 6 | 8.3 | 1.8 | | | | |
| Apogonidae | | | | | | | | | |
| *Foa brachygramma* | B | | 5 | 5.6 | 1.5 | 2 | | 6.3 | 0.4 |
| *Pristiapogon kallopterus* | B | | 1 | 2.8 | 0.3 | 30 | | 6.3 | 5.9 |
| Carangidae | | | | | | | | | |
| *Caranx melampygus* | | | 1 | 2.8 | 0.3 | | | | |
| Lutjanidae | | | | | | | | | |
| *Aprion virescens* | | | | | | | 1 | 6.3 | 0.2 |
| Mulllidae | | | | | | | | | |
| *Mulloidichthys spp.* | B | | | | | | 1 | 6.3 | 0.2 |
| Scorpaenidae | | | | | | | | | |
| *Pterois sphex* | E | | | | | 1 | | 6.3 | 0.2 |
| *Scorpaenopsis diabolus* | B | | 1 | 2.8 | 0.3 | | | | |
| Chaetodontidae | | | | | | | | | |
| *Chaetodon miliaris* | E | | | | | | 1 | 6.3 | 0.2 |
| *Heniochus diphreutes* | | | | | | | 1 | 6.3 | 0.2 |
| Pomacanthidae | | | | | | | | | |
| *Centropyge fisheri* | E | | 2 | 2.8 | 0.6 | | | | |
| Pomacentridae | | | | | | | | | |
| *Dascyllus albisella* | E | | | | | 2 | 50 | 12.5 | 10.2 |
| Labridae | | | | | | | | | |
| *Cheilio inermis* | | 1 | 7 | 5.6 | 2.4 | 2 | | 6.3 | 0.4 |
| *Cymolutes lecluse* | E, B | 1 | 8 | 11.1 | 2.7 | | | | |
| *Cymolutes praetextatus* | B | | 10 | 13.9 | 3.0 | | | | |
| *Iniistius baldwini* | B | | | | | | 1 | 6.3 | 0.2 |
| *Iniistius umbrilatus* | E, B | | | | | | 1 | 6.3 | 0.2 |
| *Novaculichthys taeniourus* | B | | 7 | 5.6 | 2.1 | | | | |

**Table 2** (*continued*)

| Species | E, B | Meadow canopy | | | | Sand/Rubble | | | |
|---|---|---|---|---|---|---|---|---|---|
| | | CO(N) | V(N) | % Occ | % RA | CO(N) | V(N) | % Occ | % RA |
| *Oxycheilinus bimaculatus* | | 10 | 104 | 41.7 | 34.1 | 5 | | 12.5 | 1.0 |
| *Pseudojuloides cerasinus* | | 2 | 24 | 16.7 | 7.8 | 1 | 13 | 12.5 | 2.7 |
| *Pseudocheilinus evanidus* | | | | | | | 1 | 6.3 | 0.2 |
| *Pseudocheilinus tetrataenia* | | | | | | | 1 | 6.3 | 0.2 |
| *Stethojulis balteata* | E, B | | 9 | 13.9 | 1.2 | | | | |
| Callionymidae | | | | | | | | | |
| *Callionymus decoratus* | E, B | | 2 | 2.8 | 0.6 | | | | |
| *Synchiropus corallinus* | B | | | | | 2 | | 6.3 | 0.4 |
| *Synchiropus rosulentus* | E, B | | | | | 6 | | 6.3 | 1.2 |
| *Synchiropus spp.* | B | | | | | 1 | 79 | 12.5 | 15.7 |
| Pinguipedidae | | | | | | | | | |
| *Parapercis schauinslandii* | B | | 27 | 8.3 | 8.1 | 4 | 2 | 25.0 | 1.2 |
| Gobiidae | | | | | | | | | |
| *Eviota susanae* | E, B | | | | | 1 | | 6.3 | 0.2 |
| *Gnatholepis spp.* | B | 21 | 1 | 33.3 | 6.6 | 13 | 1 | 37.5 | 2.7 |
| *Opua nephodes* | E, B | 1 | 28 | 11.1 | 8.7 | 15 | 151 | 31.3 | 32.5 |
| *Priolepis eugenius* | E, B | | | | | 1 | | 6.3 | 0.2 |
| *Priolepis farcimen* | E, B | | | | | 1 | | 6.3 | 0.2 |
| *Psilogobius mainlandi* | E, B | 2 | | 5.6 | 0.6 | 11 | 96 | 31.3 | 21.0 |
| Microdesmidae | | | | | | | | | |
| *Gunnelichthys curiosus* | B | | 4 | 8.3 | 1.2 | | 2 | 6.3 | 0.4 |
| Acanthuridae | | | | | | | | | |
| *Acanthurus blochii* | | | 3 | 2.8 | 0.9 | | | | |
| Bothidae | | | | | | | | | |
| *Bothus pantherinus* | B | | 4 | 11.1 | 1.2 | | 5 | 25.0 | 1.0 |
| Balistidae | | | | | | | | | |
| *Rhinecanthus aculeatus* | B | | 2 | 2.8 | 0.6 | | | | |
| Monacanthidae | | | | | | | | | |
| *Aluterus scriptus* | | | 1 | 2.8 | 0.3 | | | | |
| Ostraciidae | | | | | | | | | |
| *Ostracion meleagris* | | | 2 | 5.6 | 0.6 | | | | |
| Tetraodontidae | | | | | | | | | |
| *Arothron hispidus* | B | | 4 | 5.6 | 1.2 | | | | |
| *Canthigaster coronata* | B | | 2 | 5.6 | 0.6 | | | | |
| *Canthigaster jactator* | E, B | | 23 | 19.4 | 6.9 | | 3 | 6.3 | 0.6 |
| Diodontidae | | | | | | | | | |
| *Diodon hystrix* | B | | 1 | 2.8 | 0.3 | | | | |

**Table 3  Checklist of fishes found in association with deep-water *Avrainvillea* sp. meadows based on clove-oil collections and visual surveys.** All abbreviations follow Table 2.

| Species | E, B | Meadow canopy | | | | Sand/Rubble | | | |
|---|---|---|---|---|---|---|---|---|---|
| | | CO(N) | V(N) | % Occ | % RA | CO(N) | V(N) | % Occ | % RA |
| Muraenidae | | | | | | | | | |
| *Gymnothorax spp.* | | 1 | 1 | 20.0 | 2.2 | | | | |
| Serranidae | | | | | | | | | |
| *Plectranthias nanus* | B | | | | | 1 | | 33.3 | 2.3 |
| *Pseudanthias bicolor* | | | | | | 2 | | 66.7 | 4.7 |
| Apogonidae | | | | | | | | | |
| *Apogonichthys perdix* | | 9 | | 60.0 | 10.0 | 2 | | 33.3 | 4.7 |
| Carangidae | | | | | | | | | |
| *Caranx lugubris* | | | 1 | 10.0 | 1.1 | | | | |
| Lutjanidae | | | | | | | | | |
| *Aprion virescens* | | | 1 | 10.0 | 1.1 | | | | |
| Mulllidae | | | | | | | | | |
| *Parupeneus multifasciatus* | B | | | | | | 5 | 33.3 | 11.6 |
| Scorpaenidae | | | | | | | | | |
| *Iracundus signifer* | B | | | | | 2 | | 66.7 | 4.7 |
| *Sebastapistes fowleri* | B | 1 | | 10.0 | 1.1 | 2 | | 66.7 | 4.7 |
| Chaetodontidae | | | | | | | | | |
| *Chaetodon kleinii* | | | | | | 1 | | 33.3 | 2.3 |
| Pomacentridae | | | | | | | | | |
| *Chromis leucura* | | | | | | 1 | | 33.3 | 2.3 |
| Labridae | | | | | | | | | |
| *Bodianus bilunulatus albotaeniatus* | E | | 1 | 10.0 | 1.1 | | | | |
| *Cirrhilabrus jordani* | E | | 21 | 10.0 | 23.3 | | | | |
| *Oxycheilinus bimaculatus* | | 4 | | 30.0 | 4.4 | 2 | 3 | 100.0 | 11.6 |
| *Pseudocheilinus evanidus* | | 3 | | 30.0 | 3.3 | | 4 | 33.3 | 9.3 |
| *Pseudocheilinus octotaenia* | | | | | | 1 | | 33.3 | 2.3 |
| *Pseudojuloides cerasinus* | | 1 | 14 | 20.0 | 16.7 | | | | |
| Callionymidae | | | | | | | | | |
| *Synchiropus corallinus* | B | | | | | 1 | | 33.3 | 2.3 |
| Pinguipedidae | | | | | | | | | |
| *Parapercis schauinslandii* | B | | | | | 6 | | 66.7 | 14.0 |
| Gobiidae | | | | | | | | | |
| *Gnatholepis spp.* | B | | | | | | 2 | 33.3 | 4.7 |
| Acanthuridae | | | | | | | | | |
| *Naso caesius* | | | 30 | 10.0 | 33.3 | | | | |
| Bothidae | | | | | | | | | |
| *Bothus pantherinus* | B | | | | | | 1 | 33.3 | 2.3 |
| Soleidae | | | | | | | | | |
| *Aseraggodes borehami* | E, B | | | | | 1 | | 33.3 | 2.3 |
| Balistidae | | | | | | | | | |

| Species | E, B | Meadow canopy | | | | Sand/Rubble | | | |
| | | CO(N) | V(N) | % Occ | % RA | CO(N) | V(N) | % Occ | % RA |
|---|---|---|---|---|---|---|---|---|---|
| *Xanthichthys auromarginatus* | | | 1 | 10.0 | 1.1 | | | | |
| Monacanthidae | | | | | | | | | |
| *Cantherhines dumerilii* | B | | | | | | 2 | 33.3 | 4.7 |
| *Aluterus scriptus* | | | | | | | 3 | 33.3 | 7.0 |
| Tetraodontidae | | | | | | | | | |
| *Arothron hispidus* | B | | 1 | 10.0 | 1.1 | | | | |
| *Canthigaster coronata* | B | | | | | | 1 | 33.3 | 2.3 |

frequently-occurring species was the wrasse *O. bimaculatus*, followed by the sandperch *Parapercis schauinslandii*, which was also the most abundant species.

## Abundance, species richness, and diversity of epi-benthic fishes from clove oil anesthetic collections

A total of 105 individuals from 19 species were collected from *H. kanaloana* habitats using clove oil anesthetic (Table 4). The eyebar goby, *Gnatholepis anjerensis* and wrasse *O. bimaculatus*, were the most abundant and frequently-collected species in *H. kanaloana* canopy. In contrast, the cloud goby, *Opua nephodes*, and Hawaiian shrimp goby *Psilogobius mainlandi* were the most abundant fishes in in open sand, and were rarely collected from meadow canopy.

The median abundance and density of fishes was significantly higher in open sand ($N_x = 4.00$ individuals/collection, $D_x = 2.67$ fish per m$^2$) when compared to the meadow canopy ($N_x = 1.00$, $D_x = 0.67$; Chi-Square $= 13.41$, $DF = 1$, $P = 0.000$ for both tests). Total species richness was also nominally higher in open sand (15 vs. 10 species); however, median species richness did not differ significantly between sub-habitat types (Chi-Square $= 0.63$, $DF = 1$, $P = 0.426$). We calculated Shannon Weiner Diversity Index values of 2.2474 and 1.5013, respectively for open sand and canopy.

A total 16 species and 42 individuals were collected from *Avrainvillea* sp. sub-habitats using clove oil anesthetic (Table 5). The cardinalfish *Apogonichthys perdix* was the most abundant species collected from *Avrainvillea* sp. canopy whereas the sandperch *Parapercis schauinslandii* was most abundant species collected in open sand.

As with collections made in *H. kanaloana* meadows, fishes from *Avrainvillea* sp. collections were more abundant and densely-distributed within open sand ($N_x = 7.00$ individuals/collection, $D_x = 4.67$ fish per m$^2$) when compared to meadow canopy ($N_x = 2.50$, $D_x = 1.67$; Chi-Square $= 11.00$, $DF = 1$, $P = 0.001$ for both tests). Median Species Richness was also significantly greater in open sand ($S_x = 6.00$ species/collection) when compared to meadow canopy ($S_x = 2.00$; Chi-Square $= 11.00$, $DF = 1$, $P = 0.001$). The Shannon Weiner Diversity Index was likewise higher in open sand ($H' = 2.3667$) vs. meadow canopy ($H' = 1.4383$).

**Table 4** **Abundance and density of fishes from 1.5 m clove oil collections within *Halimeda kanaloana* canopy ($n = 28$) and surrounding sand & rubble ($n = 12$) sub-habitats.** Median abundance (fish per 1.5 m collection, $N$), species richness ($S$), and a Shannon–Wiener diversity index ($H'$) are included at bottom. All other abbreviations follow Table 2. Note that the abundance of vagile species (*) may be underestimated as these species tend to swim away when the tarp is being deployed.

| Species | E, B | Meadow (N) | Meadow density (fish per m²) | Sand (N) | Sand Density (fish per m²) |
|---|---|---|---|---|---|
| Congridae | | | | | |
| *Conger cinereus** | | 1 | 0.0238 | 0 | 0 |
| Apogonidae | | | | | |
| *Foa brachygramma** | B | 0 | 0 | 2 | 0.1111 |
| Scorpaenidae | | | | | |
| *Unidentified** | | 1 | 0.0238 | 0 | 0 |
| *Pterois sphex** | E | 0 | 0 | 1 | 0.0556 |
| Pomacentridae | | | | | |
| *Dascyllus albisella** | E | 0 | 0 | 2 | 0.1111 |
| Labridae | | | | | |
| *Cheilio inermis** | | 1 | 0.0238 | 0 | 0 |
| *Cymolutes lecluse** | E, B | 1 | 0.0238 | 0 | 0 |
| *Oxycheilinus bimaculatus** | | 10 | 0.2381 | 5 | 0.2778 |
| *Pseudojuloides cerasinus** | | 2 | 0.0476 | 1 | 0.0556 |
| Callionymidae | | | | | |
| *Synchiropus corallinus* | B | 0 | 0 | 2 | 0.1111 |
| *Synchiropus rosulentus* | E, B | 0 | 0 | 7 | 0.3889 |
| Pinguipedidae | | | | | |
| *Parapercis schauinslandii** | B | 0 | 0 | 4 | 0.2222 |
| Gobiidae | | | | | |
| *Eviota susanae* | B, E | 0 | 0 | 1 | 0.0556 |
| *Gnatholepis anjerensis* | B | 20 | 0.4762 | 12 | 0.6667 |
| *Gnatholepis cauerensis* | E, B | 1 | 0.0238 | 1 | 0.0556 |
| *Opua nephodes* | E, B | 1 | 0.0238 | 15 | 0.8333 |
| *Priolepis eugenius* | E, B | 0 | 0 | 1 | 0.0556 |
| *Priolepis farcimen* | E, B | 0 | 0 | 1 | 0.0556 |
| *Psilogobius mainlandi* | E, B | 1 | 0.0238 | 11 | 0.6111 |
| Median Abundance ($N_x$) | | 1 | | 4 | |
| Median Species Richness ($S_x$) | | 1 | | 1 | |
| Total Species Richness (S) | | 10 | | 15 | |
| Diversity ($H'$) | | 1.5013 | | 2.2474 | |
| Average Species Density m² | | | 0.0489 | | 0.1930 |
| Median Density of Fishes m² | | | 0.6667 | | 2.6667 |
| Average Density of Fishes m² | | | 0.9286 | | 3.6667 |

**Table 5  Abundance and density of fishes from 1.5 m clove oil collections within *Avrainvillea* sp. canopy ($n = 8$) and surrounding sand & rubble ($n = 3$) sub-habitats.** Median abundance (fish per 1.5 m collection, $N$), species richness ($S$), and a Shannon–Wiener diversity index ($H'$) are included at bottom. All other abbreviations follow Table 2. Note that the abundance of vagile species (*) may be underestimated as these species tend to swim away when the tarp is being deployed.

| Species | E, B | Meadow (N) | Meadow density (fish per m²) | Sand (N) | Sand density (fish per m²) |
|---|---|---|---|---|---|
| Muraenidae | | | | | |
| *Gymnothorax spp.* | | 1 | 0.0833 | 0 | 0 |
| Serranidae | | | | | |
| *Plectranthias nanus* | B | 0 | 0 | 1 | 0.2222 |
| *Pseudanthias bicolor* | | 0 | 0 | 2 | 0.4444 |
| Apogonidae | | | | | |
| *Apogonichthys perdix* | | 9 | 0.7500 | 2 | 0.4444 |
| Scorpaenidae | | | | | |
| *Iracundus signifer* | B | 0 | 0 | 2 | 0.4444 |
| *Scorpaenopsis fowleri* | B | 1 | 0.0833 | 2 | 0.4444 |
| *Unident* | | 0 | 0 | 1 | 0.2222 |
| Chaetodontidae | | | | | |
| *Chaetodon kleinii* | | 0 | 0 | 1 | 0.2222 |
| Pomacentridae | | | | | |
| *Chromis leucura* | | 0 | 0 | 1 | 0.2222 |
| Labridae | | | | | |
| *Oxycheilinus bimaculatus* | | 4 | 0.3333 | 2 | 0.4444 |
| *Pseudocheilinus evanidus* | | 3 | 0.2500 | 0 | 0 |
| *Pseudocheilinus octotaenia* | | 0 | 0 | 1 | 0.2222 |
| *Pseudojuloides cerasinus* | | 1 | 0.0833 | 0 | 0 |
| Callionymidae | | | | | |
| *Synchiropus corallinus* | B | 0 | 0 | 1 | 0.2222 |
| Pinguipedidae | | | | | |
| *Parapercis schauinslandii* | B | 0 | 0 | 6 | 1.3333 |
| Soleidae | | | | | |
| *Aseraggodes borehami* | E, B | 0 | 0 | 1 | 0.2222 |
| Median Abundance ($N_x$) | | 2.50 | | 7.00 | |
| Median Species Richness ($S_x$) | | 2 | | 6 | |
| Total Species Richness ($S$) | | 6 | | 13 | |
| Diversity ($H'$) | | 1.4383 | | 2.3667 | |
| Average Species Density m² | | | 0.0990 | | 0.3194 |
| Median Density of Fishes m² | | | 1.6667 | | 4.6667 |
| Average Density of Fishes m² | | | 1.5833 | | 5.1111 |

## DISCUSSION

Deep-water *Avrainvillea* sp. and *H. kanaloana* meadows form complex three-dimensional habitats in an otherwise two-dimensional sandy environments. This structure provides habitat and shelter for numerous fish species. Of the two habitat types, *H. kanaloana* was most diverse supporting a total of 49 species from 25 fish families. In contrast, *Avrainvillea* sp. meadows contained 28 species from 19 families, though it is likely that the lower numbers for this species may be due to the smaller sample size for this species. Within-canopy diversity ($H'$) was nominally greater for *H. kanaloana* (2.52) than *Avrainvillea* sp. (1.87). In comparison, *Friedlander & Parrish (1998)* estimated that fish diversity on a shallow Kauai reef ranged between 1.72 and 2.54. Therefore, it would appear that the diversity of these deep-water macroalgal meadows is similar to that of shallow Hawai'i reefs.

Wrasses (Family Labridae) were the most speciose taxon in both habitats (11 and six species, respectively), followed by gobies for *H. kanaloana* (six species). The wrasse *Oxycheilinus bimaculatus* and cardinalfish *Apogonichthys perdix* were the most frequently-occurring species within the *H. kanaloana* and *Avrainvillea* sp. canopies, respectively. These species were considerably less common and abundant in open sand, indicating a strong habitat preference for the meadow canopy. Other common species that showed strong associations with meadow canopy include the toby, *Canthigaster jactator* (*H. kanaloana*) and the wrasses *Cymolutes lecluse* and *C. praetextatus* (*H. kanaloana*) and *Pseudojuloides cerasinus* (both species). With the exception of *A. perdix*, which is nocturnal, we observed each of these species actively foraging within the meadow canopy on numerous occasions. Thus, meadow canopy appears to be an essential habitat for these species.

Most of the other common species were epibenthic sand-dwellers (e.g., gobies, dragonettes, sandperch, and flatfishes). Many of these species also occur in sandy areas near coral reefs (see *Greenfield, 2003*). In most cases, these species were actually more abundant in open sand rather than canopy (Tables 2 and 3), suggesting that their association with the meadows is incidental rather than a result of any inherent habitat preference.

### Percent endemism

We estimate the percent endemism for fishes living in *H. kanaloana* and *Avrainvillea* sp. habitats (including adjacent sandy areas) to be 28.6% and 10.7%, respectively. Given that approximately 25% of Hawaii's fish species are considered to be endemic (*Randall, 2007*), *H. kanaloana* meadows harbor a slightly greater proportion of endemic species than would be expected by chance. In comparison, approximately 46% of fishes surveyed within mesophotic depths in the NWHI are endemic (*Kane, Kosaki & Wagner, 2014*), and endemism may reach 100% in some areas (*Kosaki et al., 2016*). Thus, compared to mesophotic habitats in the NWHI, these macroalgal meadows have proportionally fewer endemic species. We speculate that the nominally higher endemism documented for *H. kanaloana* habitats could be a result of coevolution; *H. kanaloana* is native to Hawai'i, with the largest meadows reported from the Maui-Nui complex (*Spalding, 2012*). It is possible that some of these endemic fish species may have evolved a commensal relationship with the alga. Although only five of the 14 endemic species regularly reside in *H. kanaloana* meadows, it is possible that some of the sand-dwelling endemics may occasionally dart

into the canopy to avoid predators or may opportunistically exploit the new habitat created when blowouts (barren areas) are formed by the scour of winter swells. In contrast, *Avrainvillea* sp. is an invasive species which was first reported off Kahe Point, Oʻahu in 1981 (*Brostoff, 1989*). This species supports only three endemic species, with two (*C. jordani* and *Bodianus albotaeniatus*) recorded from canopy and one (*Aseraggodes borehami*) recorded from open sand. Given its supposed recent arrival, it is possible that endemic fish species have had less time to adapt to this unique habitat.

## Abundance of Herbivores and Food fish

Contrary to our hypothesis that *H. kanaloana* and *Avrainvillea* sp. meadows may serve as foraging grounds for herbivorous fish species, we found that obligate herbivores were quite rare in both meadow types and were represented by only two species: the angelfish *Centropyge fisheri* (*H. kanaloana)* and the surgeonfish *Acanthurus blochii* (*Avrainvillea* sp.). Neither species was common or abundant, nor do they feed on macroalgae. *Centropyge fisheri* is an obligate herbivore (*Thresher & Colin, 1986*)  that likely feeds on turf algae, whereas *A. blochii* typically feeds primarily on benthic algae on reefs or algal films covering sand (*Randall, 2005*). Surprisingly, parrotfishes (Family Scaridae) and Blennies (Family Bleniidae) were completely absent from the deepwater meadows, despite the fact that we have observed them in shallow-water (<1 m) *Avrainvillea* sp. meadows (R Langston, pers. obs., 2006). Six species (*Rhinecanthus aculeatus, Aluterus scriptus, Ostracion meleagris, Arothron hispidus, Canthigaster jactator*, and *C. coronata*) from *H. kanaloana* meadows and two from *Avrainvillea* sp. meadows (*A. hispidus* and *C. coronata*) are reported to be omnivorous, and occasionally consume algae. However, given their small size and limited abundance, it is unlikely that they consume significant amounts of macroalgal biomass. These results corroborate the work of *Spalding (2012)* who found little evidence of feeding scars on plants collected from deep-water *H. kanaloana* meadows. The absence of herbivores in these habitats may, in part, be due to their extreme depths. Several studies (e.g., *Brokovich et al., 2010*; *Fukunaga et al., 2016*; *Larkum, Drew & Crossett, 1967*; *Thresher & Colin, 1986*) report that herbivorous fish species are rare below 30 m. In addition, the lack of herbivory may be due to the low digestibility of the algae; both *Avrainvillea* sp. and *Halimeda sp.* contain numerous compounds which may deter herbivory (*Meyer et al., 1994*; *Hay et al., 1990*; *Paul & Alstyne, 1992*). In addition, the partial calcification of *H. kanaloana* may serve as an added impediment to herbivory (*Schupp & Paul, 1994*).

Commercially- or recreationally-important food-fish species were likewise rare in surveys of either meadow habitat. Three species each were recorded from *H. kanaloana* (*Caranx melampygus, Aprion virescens*, and *Mulloidichthys* sp.) and *Avrainvillea* sp. habitats (*Caranx lugubris, Parupeneus multifasciatus,* and *A. virescens*), however, each of these records was based on a single individual (Tables 2 and 3). In contrast, our surveys do indicate that deep-water *Avrainvillea* sp. meadows may be an important habitat for the flame wrasse, *C. jordani*, which is important in the Hawaiʻi aquarium fish trade. According to *Walsh et al. (2003)*, commercial fishers in the state reported catches of 13,919 *C. jordani* between the years 1976 and 2003. They estimated total wholesale value for the catch to be $133,116. Based on our review of two online fish sellers (liveaquaria.com and

petsolutions.com- July 2016), *C. jordani* retails for $130 (juvenile female) to $300 (adult male) each. Assuming the extraordinary price of *C. jordani* is a result of high demand among aquarium enthusiasts, it is possible that aquarium fishers may eventually target deep-water *Avrainvillea* sp. meadows as a potential source for this species.

## Abundance, species richness, and diversity of epi-benthic fishes from clove oil anesthetic collections

Our clove oil surveys detected significantly higher abundances and densities of epibenthic fishes in open sand when compared to meadow canopy for both *H. kanaloana* and *Avrainvillea* sp. Median species richness was also significantly higher in sand vs. meadow canopy for *Avrainvillea* sp. , and diversity ($H'$) was nominally higher in open sand for both species. These results are surprising given that several studies (e.g., *Chittaro, 2004*; *Omena & Creed, 2004*; *Ornellas & Coutinho, 1998*) have documented that diversity and abundance of fishes and invertebrates are usually highest within meadow canopy. Moreover, two recent surveys of the invertebrate fauna of *H. kanaloana* and *Avrainvillea* sp. meadows further support this hypothesis. *Magalhaes & Bailey-Brock (2014)* found that infaunal polychaetes were considerably more abundant and diverse in shallow *Avrainvillea* sp. meadows, when compared to adjacent sandy areas. *Fukunaga (2008)* likewise found that polychaetes were more abundant and diverse within *H. kanaloana* canopy. She also reported that epibenthic invertebrates were more diverse and speciose within meadows, but that abundance did not differ significantly between the two sub-habitats.

It is possible that the greater abundance of fishes in open sand may, in part, reflect a sampling bias. Fish inhabiting dense canopy are more likely to be overlooked by visual surveys when compared to individuals occurring on bare sand. Similarly, fish anesthetized in clove oil stations are more easily recovered in bare sediment than in dense canopy (this is particularly true for *Avrainvillea* sp. meadows given their dense, bladed morphology). Alternatively, it is possible that the greater abundance of some species on open sand may be related to differences in habitat preference or sediment composition between vegetated and non-vegetated areas. For example, within *Halimeda kanaloana* meadows, the Hawaiian shrimp goby, *P. mainlandi* was collected almost exclusively on bare sediment. This species lives in burrows constructed by the snapping shrimps, *Alpheus rapax* and *A. rapicida* (*Randall, 2007*). In a concomitant study of the epibenthic invertebrates from the same *H. kanaloana* meadows, *Fukunaga (2008)* recorded a total of 131 *A. rapax* in sand patches and only three individuals within the meadow canopy. Given this, it is not surprising that *P. mainlandi* was also rare within meadow canopy; without the presence of its invertebrate symbiont, the goby would have a difficult time finding shelter. We believe that the dearth of *A. rapax* burrows within meadow canopy may be related to sub-surface algal growth. Each *H. kanaloana* plant has a large, bulbous holdfast over 8 cm in length that penetrates deeply into the substrate, forming a network of stringy rhizoids that extend out into the surrounding sediment (*Verbruggen et al., 2006*). In addition, most *H. kanaloana* meadows contain several hundred plants per m$^{-2}$ (*Spalding, 2012*), thus forming a dense concentration of rhizoids and holdfasts in the sediment that may be difficult for burrow-forming species, such as *A. rapax*, to penetrate. In contrast, *Avrainvillea* sp. has a dense and

spongy holdfast that forms a terraced, penetrating mat over the sediment (*Huisman, Abbott & Smith, 2007*). These mats sequester fine sediments under their holdfast, forming anoxic mounds of sediment (*Littler, Littler & Brooks, 2005*). It is possible that these structures may likewise negatively impact benthic-dwelling and bioturbator speces. Other published studies support this hypothesis. For example, *Milazzo et al. (2004)* found that numbers of Bucchich's goby, *Gobius bucchichi* increased significantly in plots where the algal biomass was experimentally reduced. This species feeds primarily on benthic mollusks and prefers open sandy habitats (*Fasola et al., 1997*). Thus, it is not surprising that its abundance would increase when more suitable habitat was made available through the algal removal experiments. *Neira, Levin & Grosholz (2005)* likewise documented a similar shift in the invertebrate macrofauna for tidal flats invaded by a hybrid cordgrass. They found that the densities of macroinvertebrates were 75% lower in the vegetated flats, when compared to un-vegetated areas, and that species richness was also lower in the vegetated areas. Although we did not experimentally manipulate the algal biomass in this study, it does appear that benthic-dwelling and bioturbator species are numerically more abundant in open sand. Within *H. kanaloana* habitats (Table 4), these species accounted for 86% of individuals collected within open sand vs. 61% collected meadow canopy. A similar relationship is evident for *Avrainvillea* sp. collections (Table 5); 57% of individuals collected from open sand were from benthic-dwelling or bioturbator species, whereas only five percent of those collected from canopy were classified as such. Thus, we suggest that the presence of surface- and subsurface algal growths may negatively impact the abundance and diversity of small epi-benthic fishes by reducing the amount of favorable habitat and feeding areas available to these species.

## Advantage of clove oil collections

The use of clove oil anesthetic greatly enhanced our ability to estimate the fish diversity within *H. kanaloana* and *Avrainvillea* sp. meadows. Of the 65 unique species recorded in this study, 16 (25%) were detected only in clove oil stations alone. Thus, the use of clove oil anesthetic increased our overall estimate of species richness by 32.7%. It also enabled us to more accurately estimate the abundance and species richness of small-bodied fishes (gobies, scorpion fishes, cardinalfishes, dragonettes, and small wrasses), many of which are difficult to identify to species without the use of a dissecting scope. In some cases, species collected with clove oil but missed by visual surveys were also quite common. For instance, the cardinalfish *Apogonichthys perdix* was found to be the most frequently-occurring species in *Avrainvillea* sp. canopy (Table 3) though, due to its cryptic coloration and nocturnal nature, it was never observed in visual surveys. In a related study, *Fukunaga (2008)* measured the diversity and abundance of epibenthic invertebrates within *H. kanaloana* meadows using both visual surveys and clove-oil collections. She recorded 15 species of invertebrates in the visual surveys and 20 additional species in the clove-oil surveys. In this case, the use of clove oil anesthetic increased her ability to detect small epibenthic invertebrates by 133%. Together, these data highlight the utility of anesthetics (or ichthyocides) in estimating the diversity and abundance of small-bodied fishes and epibenthic invertebrates.

## ACKNOWLEDGEMENTS

We are grateful to D Pence, A Fukunaga, S Hau, K Peyton, and C Stobeneau for assistance with field surveys and logistical support. Small boat support was kindly provided by the Hawai'i Department of Aquatic Resources, Department of Land and Natural Resources on Maui.

### Funding

This work was supported by the National Oceanic and Atmospheric Administration (NOAA) National Undersea Research Program's Hawai'i Undersea Research Laboratory, NOAA Sponsored Coastal Ocean Science Hawai'i Coral Reef Initiative program to the University of Hawai'i (NA05NOS4261157), and by a grant to CM Smith (NA03NOS4780020) for resources on Maui. The funders had no role in study design, data collection and analysis, decision to publish, or preparation of the manuscript.

### Grant Disclosures

The following grant information was disclosed by the authors:
National Oceanic and Atmospheric Administration (NOAA).
National Undersea Research Program's Hawai'i Undersea Research Laboratory.
Coastal Ocean Science Hawai'i Coral Reef Initiative program: NA05NOS4261157, NA03NOS4780020.

### Competing Interests

The authors declare there are no competing interests.

### Author Contributions

- Ross C. Langston conceived and designed the experiments, performed the experiments, analyzed the data, contributed reagents/materials/analysis tools, wrote the paper, prepared figures and/or tables, reviewed drafts of the paper.
- Heather L. Spalding contributed reagents/materials/analysis tools, wrote the paper, prepared figures and/or tables, reviewed drafts of the paper.

### Animal Ethics

The following information was supplied relating to ethical approvals (i.e., approving body and any reference numbers):

These research activities in this paper were carried out under State of Hawai'i Department of Land and Natural Resources permits PRO-2006-28 and PRO-2007-47 and University of Hawai'i Institutional Animal Care and Use Committee (IACUC) protocol # 06-058.

### Field Study Permissions

The following information was supplied relating to field study approvals (i.e., approving body and any reference numbers):

Permission to use clove oil anesthetic for benthic fish collections was granted under permits # PRO 2006-28 and PRO 2007-47: Permit to engage in certain prohibited activities in the State of Hawai'i.

## Data Availability

The raw data has been supplied as Supplemental Information 1.

## Supplemental Information

Supplemental information for this article can be found online at http://dx.doi.org/10.7717/peerj.3307#supplemental-information.

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
