# Peer review of "A survey of fishes associated with Hawaiian deep-water Halimeda kanaloana (Bryopsidales: Halimedaceae) and Avrainvillea sp. (Bryopsidales: Udoteaceae) meadows"

_PeerJ, doi:10.7717/peerj.3307_

## Round 0.1 · original submission · Major Revisions

Please, consider all the suggestions given by reviewers in the revised manuscript. Furthermore, the English should be corrected by a native speaker.

Reviewer 1 ·

Basic reporting

Authors surveyed the fish assemblages between two meadow habitats and surrounding rubble/sandy area to illustrate the fish inhabitants of meadows in mesophotic ecosystems.
Generally, the finding was novel, I believe author- HS is an expert of deep water macroalgae in this region. The mesophotic research involved difficult technical diving skill (CCR) in a high-risk environment. I can see the value of this study in the region as well as on a global scale. However, I find that the writing and the structure of this paper does not meet the common requirement of scientific paper, especially the result and discussion were in too much detail but lacked interpretation.

Experimental design

I barely see an empirical exp. design to answer a hypothesis or question. Mostly it is a survey report. There is barely any statistical approach and figures provided to illustrate the findings.

Validity of the findings

Authors use the percentage occurrence and relative abundance to examine the different of fish assemblages between two meadows. Authors also listed the species in each habitat and some of them are endemic and herbivorous species. Nothing is wrong with the finding, but it is difficult for the reader to focus what the authors really want to ask.

Additional comments

Overall, the finding of this paper is novel and informative. However, from the title and written content, I found it is more likely a report to me instead of a scientific article with a hypothesis and statistical approach to examine the finding. The results were duplicated in the tables and text; this is confusing and makes it difficult for the reader to see the importance of the finding. And I did not see author provide enough background why the objectives of this paper were to compare the fishes between meadow and sandy/rubble area as well as commercial species.

·

Basic reporting

The goals of the study are clearly stated, and clearly identify a gap in our knowledge of coral reef associated communities. Discussion of potential sampling biases was handled well (Lines 262-onward) and addresses some of the concerns I was prepared to note. A figure noting the location of the study sites on two islands would be useful for those not familiar with the geography of the Hawaiian Islands.

Experimental design

The sampling methods and design are appropriate to the questions of interest, and are described in good detail. The combination of methods (visual surveys, clove oil sampling) is above and beyond the minimum necessary to address these questions, and provides and added dimension that actually constitutes a study within a study, specifically, a comparison of sampling methods.

Validity of the findings

The conclusions are stated clearly, but in this format (combined results and discussion), the discussion and citations of relevant literature are not as fully developed as they might be in a separate, stand-alone discussion section. Some suggestions are itemized below.
Sample sizes in Avrainvillea canopy and sand habitats are relatively small, and in an ideal world, a more balanced design (roughly comparable numbers of surveys in both Avrainvillea and Halimeda habitats) would be desirable. Nevertheless the results appear to be robust and support the conclusions drawn. Additionally, I recognize the challenges associated with working at depth, and I commend the authors for taking on an interesting and logistically challenging project.

Specific comments:

Line 169 (paragraph): Why do you think endemism was so much higher in Halimeda vs. Avrainvillea? Perhaps because endemic species co-evolved with Halimeda but not Avrainvillea? It’s well worth pointing out that native species appear to provide better fish habitat than introduced species. Please speculate! This is one of the most interesting findings of the paper and warrants additional discussion.

Line 174: consider citing Kosaki et al. 2016 in addition to Kane et al. 2014 in regard to high mesophotic endemism in NWHI fish assemblages.

Line 190: consider citing Fukunaga et al. 2016 on the decline in abundance of herbivores with depth in the NWHI, and Thresher and Colin 1986 for the Marshall Is.

Additional comments

A paper does not have to be lengthy or complex and convoluted to constitute a valuable contribution to its field. Not only is does this paper describe fish fauna associated with habitats adjacent to coral reef habitats, it is also a valuable comparison of two methods of assessing fish biodiversity and abundance (visual surveys vs. chemical collections). This will serve to inform future research at mesophotic depths where coral cover often decreases with depth, and alternate coral reef associated habitat types predominate. Few papers have looked at assemblages of coral reef associated fauna at mesophotic depths in non-coral, soft substrate habitats. This paper broadens the scope of research at mesophotic depths, which will only enhance our understanding these habitats as extensions of the coral reefs themselves. This definitely should be published.

---

## Round 0.2 · accepted · Accept

Congratulations for this work and thank you for improving your manuscript.

·

Basic reporting

No additional comments.

Experimental design

No additional comments.

Validity of the findings

No additional comments.

Additional comments

No additional comments.